# Genetic Divergence and Population Structure of *Xanthomonas albilineans* Strains Infecting *Saccharum* spp. Hybrid and *Saccharum officinarum*

**DOI:** 10.3390/plants12101937

**Published:** 2023-05-09

**Authors:** Zhong-Ting Hu, Mbuya Sylvain Ntambo, Jian-Ying Zhao, Talha Javed, Yang Shi, Hua-Ying Fu, Mei-Ting Huang, San-Ji Gao

**Affiliations:** 1National Engineering Research Center for Sugarcane, Fujian Agriculture and Forestry University, Fuzhou 350002, China; hzt9901@163.com (Z.-T.H.); zhaojyfafu@126.com (J.-Y.Z.); mtahaj@fafu.edu.cn (T.J.); syang24@163.com (Y.S.); mddzyfhy@163.com (H.-Y.F.); hmt159379@163.com (M.-T.H.); 2Université de Lubumbashi, Faculté des Sciences Agronomiques, Département de Phytotechnie, Laboratoire de Recherche en Biofortification, Défense et Valorisation des Cultures (BioDeV), Lubumbashi 7010, Congo; ntambos@africau.edu

**Keywords:** genetic diversity, housekeeping gene, population structure, rep-PCR, sugarcane, *Xanthomonas albilineans*

## Abstract

Leaf scald caused by *Xanthomonas albilineans* (*Xa*) is a major bacterial disease in sugarcane that represents a threat to the global sugar industry. Little is known about the population structure and genetic evolution of this pathogen. In this study, 39 *Xa* strains were collected from 6 provinces in China. Of these strains, 15 and 24 were isolated from *Saccharum* spp. hybrid and *S. officinarum* plants, respectively. Based on multilocus sequence analysis (MLSA), with five housekeeping genes, these strains were clustered into two distinct phylogenetic groups (I and II). Group I included 26 strains from 2 host plants, *Saccharum* spp. hybrid and *S. officinarum* collected from 6 provinces, while Group II consisted of 13 strains from *S. officinarum* plants in the Zhejiang province. Among the 39 *Xa* strains, nucleotide sequence identities from 5 housekeeping genes were: *ABC* (99.6–100%), *gyr*B (99.3–100%), *rpo*D (98.4–100%), *atp*D (97.0–100%), and *gln*A (97.6–100%). These strains were clustered into six groups (A–F), based on the rep-PCR fingerprinting, using primers for ERIC2, BOX A1R, and (GTG)5. UPGMA and PCoA analyses revealed that group A had the most strains (24), followed by group C with 11 strains, while there was 1 strain each in groups B and D–F. Neutral tests showed that the *Xa* population in *S. officinarum* had a trend toward population expansion. Selection pressure analysis showed purification selection on five concatenated housekeeping genes from all tested strains. Significant genetic differentiation and infrequent gene flow were found between two *Xa* populations hosted in *Saccharum* spp. hybrids and *S. officinarum*. Altogether, these results provide evidence of obvious genetic divergence and population structures among *Xa* strains from China.

## 1. Introduction

Sugarcane is an important cash crop that represents 80% of sugar and 40% of ethanol production worldwide [1]. China produced 9.56 million tons of sugar from sugarcane and sugar beets during the 2021/2022 crushing season, whereby the three main sugarcane-producing provinces (Guangxi, Yunnan, and Guangdong) accounted for 64.0%, 20.3%, and 5.7%, respectively [2]. Climate change is leading to diverse abiotic and biotic stressors that have major impacts on food production worldwide [3]. Plant diseases can occur in both the pre- and postharvest phase of crop production chains, resulting in a 13–22% annual yield loss of major food crops [4]. Leaf scald commonly occurs throughout the growth and development stage of sugarcane plants in sugarcane-planting areas worldwide. In China, sugarcane leaf scald was first recorded in Taiwan in the 1980s based on symptomatology and traditional cytological and biochemical techniques [5]. Leaf scald is caused by *X. albilineans* (*Xa*) and molecular analysis identified the presence of this disease in nine provinces of China [5,6,7].

*Xanthomonas* spp. are Gram-negative bacteria belonging to the family *Xanthomonadaceae* and cause a range of diseases in over 400 monocotyledonous and dicotyledonous plants worldwide [8,9]. Most *Xanthomonas* spp. strains are characterized by the presence of a membrane-bound pigment “Xanthomonadin” that produces a yellow color in culture media [9,10]. Each *Xanthomonas* spp., which shows a different host range or produces different disease symptoms is classified as a separate species, while some are further taxonomically classified into different subspecies and pathovars that show a high degree of host plant specificity and tissue specificity, which can involve the host plant vascular system or mesophyll tissues [9,11,12]. Taxonomical classification of different subspecies and pathovars in the *Xanthomonas* species confirms a particular adaptation to plants due to the expression of virulence factors [13]. *Xanthomonas* spp. are rapidly evolving microbes, as evidenced by the evolutionary dynamics of an emerging and variant pathovar of this bacterial species [9]. Among *Xanthomonas* spp., *Xa* is assigned to a unique clade in the phylogenetic tree based on whole-genome sequences; its genome has undergone significant erosion and possesses an “open” and flexible genome [13,14].

Substantial genetic plasticity exists among *Xa* strains around the world. A total of 28 *Xa* strains from 11 different countries can be split into 3 serovars and 6 lysovars for the pathogen [15,16]. Serological diversity of the pathogen has been confirmed using a combination of monoclonal antibodies and DNA fingerprinting of 38 *Xa* strains from various geographical locations [17]. Pulsed-field gel electrophoresis (PFGE) and multilocus sequence analysis (MLSA) revealed that 218 isolates from 31 geographical locations were clustered into 10 genetic groups (PFGE-A to PFGE-J) [18,19,20]. Our recent studies revealed that the majority of *Xa* strains in China belonged to PFGE-B [21], while some strains from a novel group were proposed based on MLSA and simple sequence repeats (SSRs) [7,22]. In addition to the MLSA and SSR methods, other fingerprinting tools have also been used to investigate *Xa* genetic divergence, including restriction fragment length polymorphism (RFLP) [18], random amplified fragment polymorphism (RAPD) [23], rep-PCR [24], and amplified fragment length polymorphism (AFLP) [25]. Recently, whole-genome sequencing of *Xa* strains was performed on two strains from China [26,27] and four strains from Brazil [28]. These strains were further used together alongside publicly available strains in the NCBI database to explore genomic diversity.

Our previous studies examined the genetic diversity of *Xa* in China based on MLSA and SSR techniques [7,21,22], although the population divergence and evolution of this pathogen remain largely unexplored. The aims of this study were to: (a) use rep-PCR and MLSA to investigate the genetic diversity of 39 *Xa* strains from China, (b) estimate the population structure of these strains, and (c) analyze the evolutionary driving force that forms *Xa* populations from a modern commercial cultivar (*Saccharum* spp. hybrid) and chewing cane (*S. officinarum*).

## 2. Results

### 2.1. Sequence Identity Analysis for Five Housekeeping Genes

Sequence identities among 39 *Xa* strains were investigated based on 5 individual housekeeping genes, including *ABC* (encodes the ABC transporter protein), *rpo*D (encodes the bacterial RNA polymerase β subunit), *gln*A (encodes a citrate synthase), *gyr*B (encodes the DNA gyrase β subunit), and *atp*D (encodes the ATP synthase β subunit). The lowest nucleotide sequence divergences were observed for *ABC* (99.6–100%) and *gyr*B (99.3–100%), followed by *rpo*D (98.4–100%), then, *gln*A (97.6–100%), and *atp*D (97.0–100%), suggesting that a low rate of nucleotide mutations was predominate in two genes (*ABC* and *gyr*B). A similar trend for sequence divergence was observed based on the amino acid sequences of the five genes (Appendix A).

### 2.2. Phylogenetic Grouping Based on MLSA

A neighbor-joining phylogenetic tree was constructed based on the concatenated sequences of the 5 housekeeping genes revealed that all 39 *Xa* strains were allocated in 2 major phylogenetic clades (I and II), while the reference strain REU209 (PFGE-J group) from Reunion Island (France) was assigned to a unique clade III (Figure 1). Clade I (PFGE-B group) included 26 strains hosted in several modern commercial cultivars and 2 chewing cane cultivars (Guangdonghuangpi and Badila) from 6 provinces. Clade II (Xa-RA) comprised 13 strains hosted in a chewing cane cultivar (Taoshangguozhe) from Ruian, Zhejiang province.

### 2.3. Genetic Diversity Based on Rep-PCR Analysis

Here, we analyzed 39 *Xa* strains in China using 3 rep-PCR primers for ERIC2, BOX A1R, and (GTG)5, which, on average, directed the amplification of 6–10 fragments ranging in size from 700 to 5000 bp. The BOX primer pair had the highest discriminatory power, as evidenced by a higher polymorphism information content (PIC) value (0.87). The ERIC and GTG primer pairs both had the same PIC value (0.83) (Table 1).

### 2.4. UPGMA and PCoA Analyses

UPGMA analysis based on the fingerprints generated from the three aforementioned primer sets showed that the genetic similarity coefficient among the 39 *Xa* strains ranged from 0.28 to 1.00. When the genetic similarity coefficient was 0.64, these strains could be divided into six groups (Figure 2). Group A was the largest with 24 strains from 6 provinces. Group C was the second-largest with 11 strains (XaCN42–XaCN44, XaCN46–XaCN51, XaCN54, and XaCN56) that originated from the city of Ruian in the Zhejiang province. The individual strain XaCN17 (Wuzhishan, Hainan province) was clustered in group B, while XaCN45 and XaCN55 (Ruian, Zhejiang province) were clustered in groups D and E, respectively, and XaCN14 (Xingyi, Guizhou province) was clustered in group F. PCoA analysis indicated that PCoA 1 and 2 accounted for 73.93% of the total variances. The classification of these 39 strains was identical to the UPGMA analysis (Appendix A).

### 2.5. Neutrality Tests and Selection Pressure Analysis

According to two different hosts, all 39 *Xa* strains were divided into 15 strains from the *Saccharum* spp. hybrid group (commercial cultivars) and 24 strains from the *S. officinarum* group (chewing cane). Based on five housekeeping genes, sequence identity analysis revealed 98.8–100% and 99.4–99.9% nucleotide identity for the two *Xa* strain populations from *Saccharum* spp. hybrid and *S. officinarum*, respectively. The amino acid identities were similarly high at 99.0–100% and 99.3–100%*,* respectively. The nucleotide diversity (π) showed that *Xa* strains from the *Saccharum* spp. hybrid group had higher genetic variation than *S. officinarum*. A neutral test showed that the Tajima’s D value for *Xa* strains from the *Saccharum* spp. hybrid was significantly less than 0, indicating that the population of these *Xa* strains exhibited a trend for population expansion. Selection pressure analysis showed that the five tandem genes from two *Xa* populations underwent purification selection during evolution, as evidenced by a ratio of non-synonymous to synonymous (dN/dS) rates that were <1 (Table 2).

### 2.6. Genetic Differentiation and Gene Flow

To further investigate the evolutionary process of these strains based on *Saccharum* spp. hybrids and *S. officinarum*, values describing genetic differentiation and gene flow were calculated. Values for Ks*, Z*, and Snn were 2.59155, 5.47487, and 0.65385, respectively, and all were statistically significant (*p* < 0.05), indicating obvious genetic differentiation of the two populations. The F_ST_ value (0.40461) was >0.33 and the Nm value (0.74) was <1, suggesting an infrequent gene flow between the two populations. These results indicated that local host isolation might be the main factor that formed the population constructs in the *Xa* strains in China (Table 3).

## 3. Discussion

Extensive genetic diversity among *Xanthomonas* spp. can be categorized at multiple levels including within populations and species [13]. The high genetic variability of *Xa* strains across the world has been investigated in several previous studies [19,21,29]. In our early observations, low genetic diversity of *X. albilineans* strains clustered with foreign strains in the PFGE-B group was reported in China [21]. These strains were isolated from the modern sugarcane cultivars (*Saccharum* spp. hybrid), grown within China. However, some *Xa* strains were collected from a local traditional chewing cane cultivar Taoshangguozhe (*S. officinarum*), collected in the city of Ruian (Zhejiang province), and were proposed to form a new phylogenetic group, Xa-RA, based on MLSA with two (*ABC* and *gyr*B) [22] or four (*rpo*D, *gln*A, *gyr*B, and *atp*D) genes [7].

Here, we further verified the two distinct phylogenetic groups of *Xa* strains that exist in China by MLSA with five housekeeping gene sequences (*ABC*, *rpo*D, *gln*A, *gyr*B, and *atp*D). The strains hosted in the cultivar Taoshangguozhe of *S. officinarum* formed a separate group, group II, which was distinct from other Chinese strains, whereas the *Xa* strains hosted in two chewing cane cultivars (Guangdonghuangpi and Badila) of *S. officinarum* along with *Xa* strains hosted in modern commercial cultivars clustered into the same group (Group I). Notably, the famous chewing cane cultivar Badila has been cultivated for more than half a century in southern China, while the cultivar Guangdonghuangpi was recently introduced into Zhejiang from Guangdong province [22]. Almost all *Xa* strains from China, including those grown in Guangdong province, clustered in the PFGE-B group [21,30]. Therefore, we propose that the spread of this pathogen might be due to the transport of sugarcane stalks across provinces in China. This possibility would explain why the *Xa* strains from Guangdonghuangpi and Badila of *S. officinarum* and modern commercial cultivars (*Saccharum* spp. hybrid) clustered into the same group (Group I). A similar phylogenetic grouping of these Chinese *Xa* strains was observed using an MLSA approach based on four housekeeping genes [7].

Subsequently, rep-PCR fingerprinting was employed to amplify highly conserved repetitive sequences that are generally distributed in the bacterial genome to disclose diversity among the strain genomes [24,31]. The advantages of rep-PCR are a simple and fast operation, rich amplification bands, and an absence of strain-specific DNA probes [24,32]. Additionally, REP, ERIC, and BOX sequences for rep-PCR are frequently used in polymorphism analysis of bacteria to evaluate strain diversity [31,33]. Similar to other techniques such as RFLP, rep-PCR is crucial in determining the diversity within bacterial *Xa* strains [19]. Additionally, rep-PCR was used to reveal the high genetic diversity and clonal population structure of *Acidovorax avenae* subsp. *avenae*, which is the causal agent of the red stripe in sugarcane [33]. Notably, based on the fingerprinting profile, the rep-PCR method possesses more discriminatory power than the MLSA method to differentiate among these *Xa* strains in this study. For example, four strains from Group I (XaCN14 and XaCN17) and II (XaCN45 and XaCN55), identified by MLSA, were separated into four unique groups by rep-PCR fingerprinting. However, MLSA generates a discrete dataset based on the nucleotide sequences of known genes and allows for the accurate calculation of genetic distances compared to rep-PCR [32].

Local host and environmental factors are important driving forces resulting in the emergence and selection of pathogens, including *Xanthomonas* spp. [13]. Obvious population-level genetic differentiation of *Xa* strains was observed between *S. officinarum* and *Saccharum* spp. hybrid groups, indicating that host isolation might be the main driving force causing the genetic differentiation of the *Xa* population in China, particularly for the *Xa* strains hosted in the local cultivar (Taoshangguozhe), in the city of Ruian in Zhejiang province. A previous study suggested that bacteria variants and the reaction of sugarcane cultivars contributed to the genetic and aggressive diversity within the *Xa* population in the state of São Paulo in Brazil [34]. The interaction between pathogens and specific hosts can cause genetic differentiation of pathogen populations, which is related to the resistance levels of different hosts [35]. A host-driven population shift occurred in *X. oryzae* pv. *oryzae* in response to the introduction of resistant plants [36]. However, cross-infection by dispersal rather than by host-driven speciation contributed to the evolution of *X. citri* subsp. *citri* pathotypes, resulting in the diversification of this pathotype [37]. It is worthy noted that no correlation was observed between the host plant and variation in pathogenicity of Chinese *Xa* strains [7]. Other biological variation and bacterial genomics between two *Xa* populations need be further identified.

The *Xa* strains in Group II (previously termed Xa-RA by Zhao et al. [7]) determined in this study may also be related to the specific niche and climate conditions of Ruian in Zhejiang province. Ecological conditions such as soil, climate, water, and other factors differ by geographical locations where sugarcane grows and these conditions can also promote genetic differentiation of pathogen populations [13]. Genetic diversity revealed by a phylogenomic tree based on 21 public genomic sequences of worldwide *Xa* strains can be grouped based on their geographic origin (American, African, or Asiatic), to some degree [28]. Recombination-driven horizontal gene transfer is an important factor that contributes to pathogen population structure and diversity across different *Xanthomonas* spp. [13]. Additionally, some factors such as effectors related to the secretion system, cell-wall degradation, lipopolysaccharides, and TonB-dependent receptors influence host specificity and bacterial pathogenicity in several *Xanthomonas* spp. [13,36]. The evolutionary driving force of host jumping or geographical isolation forming genetic populations among *Xa* strains in China needs to be verified at a genome-wide level using more strains from different hosts and geographical origins.

## 4. Materials and Methods

### 4.1. Collection of X. albilineans Strains and DNA Extraction

Thirty-nine *Xa* strains were isolated from sugarcane leaf or stalk samples exhibiting leaf scald symptoms [7,26]. These strains were distributed within six provinces (Fujian, Guangdong, Guangxi, Guizhou, Hainan, and Zhejiang) in China (Table 4). A Bacterial Genomic DNA Extraction Kit (Tiangen Biotechnology Co. Ltd., Beijing, China) was used to extract total bacterial genomic DNA. DNA samples with a working concentration of 100 ng/μL were used for the subsequent gene cloning.

### 4.2. Amplification of Five Housekeeping Genes among Strains

A total of 5 housekeeping genes (*ABC*, *rpo*D, *gln*A, *gyr*B, and *atp*D) were amplified from bacterial DNA from 39 *Xa* strains by PCR using published primers specific to *X. albilineans* [21,38]. The five genes selected for MLSA were also included. The primer sequences used to clone these gene fragments are listed in Appendix A. The PCR reaction system used a 50 μL reaction volume including 5.0 μL 10X PCR buffer, 4.0 μL 2.5 mmol/L dNTP mixture, 1.0 μL forward primer, and reverse primer (10 μmmol/L each), 0.25 μL 5 U/μL Ex Taq enzyme, 1.0 μL template DNA, and 37.75 μL sterile water. The PCR program was 35 cycles of 94 °C initial denaturation for 1 min; 94 °C denaturation for 30 s, 56 °C annealing for 1 min, 72 °C extension for 30 s, followed by a 5 min extension at 72 °C. All PCR products were purified and cloned into the pMD19-T vector. Three independent positive clones were randomly selected for nucleotide sequencing carried out by Sangon Biotech (Shanghai, China). All gene sequences have been deposited in the NCBI library, and the accession numbers are listed in Appendix A. The *ABC* gene sequences were designed during this study, while the sequences for the other four genes were reported by Zhao et al. (2022) [7].

### 4.3. Multilocus Sequence Analysis of Housekeeping Genes

The sequences of the five housekeeping genes in the *Xa* strains were concatenated to yield 4773 nucleotides (nt) for each strain. The corresponding sequence for the REU209 strain from France was used as a reference strain (GenBank accession no. JZII01000001) [20]. All 39 sequences were aligned using the ClustalW algorithm implemented in MEGA 11.0 and a phylogenetic tree was constructed using the neighbor-joining (NJ) method [39]. Bootstrap values were determined for 1000 replications. Determination of sequence identification was performed using BioEdit v7.1 software [40].

### 4.4. Rep-PCR Amplification of X. albilineans Genomes

Three rep-PCR primers ERIC2 (5′- AAGTAAGTGACTGGGGTGAGCG-3′), BOX A1R (5′-CTACGGCAAGGCGACGCTGACG-3′), and (GTG)5 (5′-GTGGTGGTGGTGGTG-3′), previously designed by Silvio et al. [24], were used. The PCR reaction was performed in a 25 µL reaction volume consisting of 0.25 μL 5 U/µL Ex Taq, 2.5 μL 10X Ex Taq PCR buffer, 2.0 μL 2.5 mmol/L dNTPs buffer, 1.0 µL 10 μmol/L primer, 1.0 μL genomic DNA, and 18.25 μL sterile water. PCR amplification conditions comprised an initial denaturation at 95 °C for 7 min, followed by 30 cycles at 94 °C for 1 min, primer annealing for 1 min at 50 °C (ERIC2), 53 °C (BOX A1R), or 40 °C ((GTG)5), and extension at 65 °C for 8 min. The final step was another extension step at 65 °C for 15 min. After amplification, 6 µL of the reaction products were detected by electrophoresis on a 1.5% agarose gel at 4 V/cm for 2 h. Gels were photographed using an Automatic Gel Imaging Analysis System (Shanghai Peiqing Science & Technology, Shanghai, China).

### 4.5. DNA Fingerprinting of Strains Based on Rep-PCR Datasets

Bands at each different position in the DNA fingerprint of 39 *Xa* strains were used as a molecular marker (a locus). All rep-PCR bands were included in the dataset and converted into a code with 1 (present) or 0 (absent). The polymorphism information content (PIC) of the loci was calculated according to the following formula:$$\mathrm{PIC}=1-\overset{n}{\underset{j=1}{\sum }}P_{ij}^{2}$$
where *P_ij_* represents the frequency of the *j*th allele for locus *i* and is summed across all alleles [41]. PIC is the ability of a marker to detect a polymorphism. A distance tree was constructed using clustering with the unweighted pair group method using the arithmetic mean (UPGMA) in the SHAN program of NTSYS-PC 2.10e software (University of Kansas, Lawrence, KS, USA). Principal coordinates analysis (PCoA) was conducted using tools on the online website https://cloud.oebiotech.cn/task/ (accessed on 28 November 2022).

### 4.6. Determination of Genetic Parameters of Populations

To further explore the degree of genetic evolution and differentiation among *Xa* strains from different host sources, DnaSP v5.0 software was used to determine the genetic parameters of populations based on tandem sequence sets of five housekeeping genes [42]. The genetic parameters for populations included nucleotide diversity (π), neutral test parameters (Tajima’s D), and selection pressure (dN/dS). Negative, neutral, and positive selections were indicated by dN/dS ratios < 1, =1, and >1, respectively. The genetic differentiation parameters for populations included Ks*, Z*, and Snn values, and probability (*p*-value) obtained by a permutation test with 1000 replicates. The gene flow parameters included F_ST_ values and Nm values. |Fst| > 0.33 or |Nm| < 1 indicates infrequent gene flow and |Fst| < 0.33 or |Nm| > 1 indicates frequent gene flow.

## 5. Conclusions

This study used MLSA and rep-PCR tools to reveal the genetic diversity and population structure of *Xa* strains hosted in *Saccharum* spp. hybrid and *S. officinarum* from China. Two distinct groups were proposed by MLSA with five housekeeping genes, while more distinct groups were proposed by UPGMA and PCoA analyses based on a rep-PCR dataset. The *Xa* population hosted in *S. officinarum* had a trend toward population expansion. Purification selection was the main evolutionary driving force on five genes among all tested strains. Obvious genetic differentiation was observed between two *Xa* populations hosted in *Saccharum* spp. hybrid and *S. officinarum*, while infrequent gene flow occurred between the two populations. Altogether, this work provides valuable information for a fundamental understanding of the biology of this pathogen. However, some gaps in information remain that require further exploration. For instance, the global distribution and evolution of this pathogen could be revealed by comparing *Xa* populations in China to other countries. A whole-genome sequencing approach should be used to gain a more comprehensive understanding of sequence diversity, virulence, and plant–pathogen interactions. Virulence factors such as effectors in diverse secretion systems in *Xa* strains might reveal host specificity as well as whether emerging strains are evolving and whether evasions of host immunity systems are occurring.

## Figures and Tables

**Figure 1 plants-12-01937-f001:**
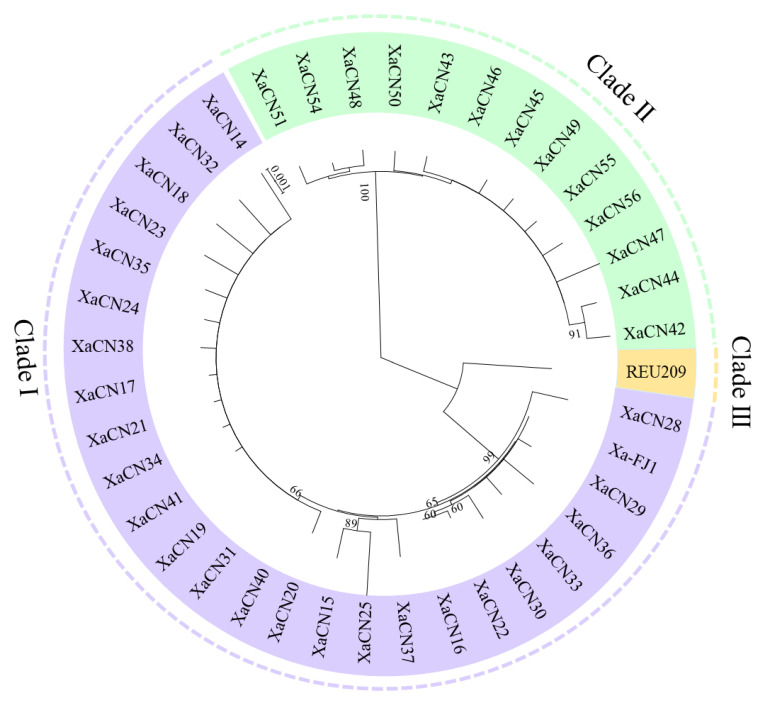
Phylogenetic tree of the concatenated sequences of 5 housekeeping genes based on 39 *X. albilineans* strains from China and a reference strain REU209 from France.

**Figure 2 plants-12-01937-f002:**
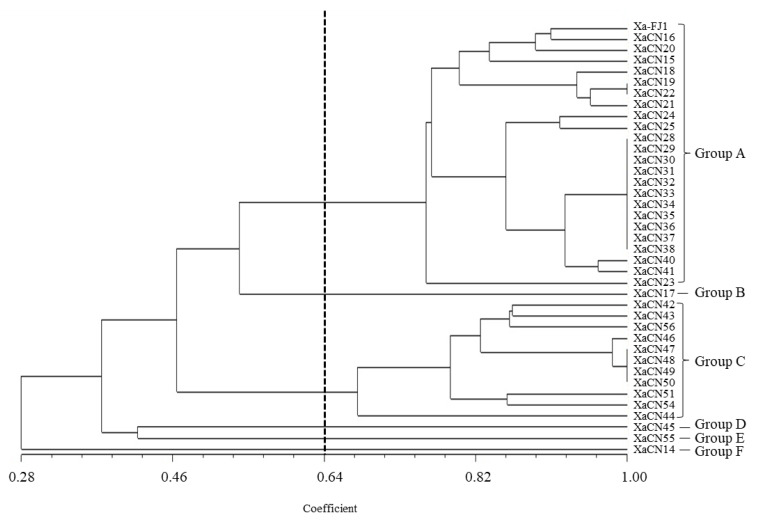
Dendrogram based on cluster analysis (UPGMA) of the estimated genetic similarity among 39 *X. albilineans* strains based on rep-PCR fingerprints.

**Table 1 plants-12-01937-t001:** DNA fingerprinting parameters for 39 *X. albilineans* strains from China based on a rep-PCR assay.

Parameter	Rep-PCR Primer for DNA Fingerprinting		
	ERIC	BOX A1R	(GTG)5
Total differentiable samples	12	15	11
Total indifferentiable samples	27	24	28
Genetic similarity	0.18–1	0.25–1	0.31–1
Total bands	24	24	27
Polymorphic bands	16	19	16
Minimum number of bands	9	13	13
Maximum number of bands	3	3	7
Average number of bands	6	9	10
Range (bp)	700–5000	900–5000	1000–5000
Polymorphism information content (PIC)	0.83	0.87	0.83

**Table 2 plants-12-01937-t002:** Genetic diversity, neutrality test, and selection pressure of the concatenated sequences for 5 housekeeping genes in 39 *X. albilineans* strains from different host sources.

Group	Identity (%)		Nucleotide Diversity (π)	Tajima’s D ^a^	dN/dS ^b^
	Nucleotide	Amino Acid			
All (*n* = 39)	98.5–100	99.0–100	0.00568	−1.04156 (ns)	0.09
*Saccharum* spp. hybrids (*n* = 15)	98.8~100	99.3–100	0.00175	−2.2732 (**)	0.78
*S. officinarum* (*n* = 24)	99.4–99.90	99.0–100	0.00605	0.09801 (ns)	0.07

^a^ **: 0.001 < *p* ≤ 0.01; ns: not significant. ^b^ dN/dS, the ratio of nonsynonymous (dN) to synonymous (dS) substitution rates.

**Table 3 plants-12-01937-t003:** Gene flow and genetic differentiation between two *X. albilineans* populations based on different hosts ^a^.

Group	Ks* (*p* Value)	Z* (*p* Value)	Snn (*p* Value)	F_st_	Nm
*Saccharum* spp. hybrids (*n* = 15) vs. *S. officinarum* (*n* = 24)	2.59155 (0.0000 ***)	5.47487 (0.0010 **)	0.65385 (0.0120 *)	0.40461	0.74

^a^ Probability (*p*-value) obtained by the permutation test (PM test) with 1000 replicates. *: 0.01 < *p* ≤ 0.05; **: 0.001 < *p* ≤ 0.01; ***: *p* ≤ 0.001.

**Table 4 plants-12-01937-t004:** Information on 39 *X. albilineans* strains from China and 1 reference strain from France.

No.	Strain ^a^	Host Variety ^b^	*Saccharum* spp.	Sampling Location	Sampling Date
1	Xa-FJ1	Yuegan 48	*Saccharum* spp. hybrid	Zhangzhou, Fujian, China	January 2015
2	XaCN14	Qiantang 8	*Saccharum* spp. hybrid	Xingyi, Guizhou, China	August 2018
3	XaCN15	Guitang 08-1589	*Saccharum* spp. hybrid	Zhanjiang, Guangdong, China	January 2018
4	XaCN16	GUC37	*Saccharum* spp. hybrid	Fusui, Guangxi, China	January 2018
5	XaCN17	ROC20	*Saccharum* spp. hybrid	Wuzhishan, Hainan, China	June 2019
6	XaCN18	Yuetang 92-1287	*Saccharum* spp. hybrid	Wuzhishan, Hainan, China	June 2019
7	XaCN19	Funong 93-23	*Saccharum* spp. hybrid	Wuzhishan, Hainan, China	June 2019
8	XaCN20	ROC20	*Saccharum* spp. hybrid	Wuzhishan, Hainan, China	June 2019
9	XaCN21	HoCP 93-750	*Saccharum* spp. hybrid	Sanya, Hainan, China	June 2019
10	XaCN22	16139	*Saccharum* spp. hybrid	Danzhou, Hainan, China	June 2019
11	XaCN23	NA	*Saccharum* spp. hybrid	Danzhou, Hainan, China	June 2019
12	XaCN24	Yunzhe 11-1204	*Saccharum* spp. hybrid	Zhanjiang, Guangdong, China	June 2019
13	XaCN25	Regan 15-285	*Saccharum* spp. hybrid	Zhanjiang, Guangdong, China	June 2019
14	XaCN28	NA	*Saccharum* spp. hybrid	Wuming, Guangxi, China	May 2019
15	XaCN29	Yuetang 93-159	*Saccharum* spp. hybrid	Fuzhou, Fujian, China	May 2019
16	XaCN30	Guangdonghaungpi	*S. officinarum*	Wenling, Zhejiang, China	June 2019
17	XaCN31	Guangdonghaungpi	*S. officinarum*	Wenling, Zhejiang, China	June 2019
18	XaCN32	Guangdonghaungpi	*S. officinarum*	Wenling, Zhejiang, China	June 2019
19	XaCN33	Guangdonghaungpi	*S. officinarum*	Wenling, Zhejiang, China	June 2019
20	XaCN34	Guangdonghaungpi	*S. officinarum*	Wenling, Zhejiang, China	June 2019
21	XaCN35	Guangdonghaungpi	*S. officinarum*	Wenling, Zhejiang, China	June 2019
22	XaCN36	Guangdonghaungpi	*S. officinarum*	Wenling, Zhejiang, China	June 2019
23	XaCN37	Badila	*S. officinarum*	Wenling, Zhejiang, China	June 2019
24	XaCN38	Guangdonghaungpi	*S. officinarum*	Wenling, Zhejiang, China	June 2019
25	XaCN40	Guangdonghaungpi	*S. officinarum*	Wenling, Zhejiang, China	June 2019
26	XaCN41	Guangdonghaungpi	*S. officinarum*	Wenling, Zhejiang, China	June 2019
27	XaCN42	Taoshanguozhe	*S. officinarum*	Ruian, Zhejiang, China	August 2019
28	XaCN43	Taoshanguozhe	*S. officinarum*	Ruian, Zhejiang, China	August 2019
29	XaCN44	Taoshanguozhe	*S. officinarum*	Ruian, Zhejiang, China	August 2019
30	XaCN45	Taoshanguozhe	*S. officinarum*	Ruian, Zhejiang, China	August 2019
31	XaCN46	Taoshanguozhe	*S. officinarum*	Ruian, Zhejiang, China	August 2019
32	XaCN47	Taoshanguozhe	*S. officinarum*	Ruian, Zhejiang, China	August 2019
33	XaCN48	Taoshanguozhe	*S. officinarum*	Ruian, Zhejiang, China	August 2019
34	XaCN49	Taoshanguozhe	*S. officinarum*	Ruian, Zhejiang, China	August 2019
35	XaCN50	Taoshanguozhe	*S. officinarum*	Ruian, Zhejiang, China	August 2019
36	XaCN51	Taoshanguozhe	*S. officinarum*	Ruian, Zhejiang, China	August 2019
37	XaCN54	Taoshanguozhe	*S. officinarum*	Ruian, Zhejiang, China	August 2019
38	XaCN55	Taoshanguozhe	*S. officinarum*	Ruian, Zhejiang, China	August 2019
39	XaCN56	Taoshanguozhe	*S. officinarum*	Ruian, Zhejiang, China	August 2019
40	REU209	NA	NA	Reunion Island, France	1995

^a^ Strain Xa-FJ1 (No. 1) and No. 2–39 were referenced by [27] and [7], respectively; strain REU209 (No. 40) was referenced by [14]. ^b^ NA: not available.

## Data Availability

All data supporting the findings of this study are available within the paper and its Appendix A.

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
