# Peer review of "Genetic Divergence and Population Structure of Xanthomonas albilineans Strains Infecting Saccharum spp. Hybrid and Saccharum officinarum"

_plants, 2023, doi:10.3390/plants12101937_

Round 1

Reviewer 1 Report

The manuscript deals with study on the population structure and genetic evolution of Xa, which is a major bacterial disease of sugarcane, in China. They collected 39 strains of Xa from six provinces in China and used multilocus sequence analysis (MLSA) and rep-PCR fingerprinting to analyze the genetic diversity and population structure of these strains. The authors also conducted neutral tests, selection pressure analysis, genetic differentiation and gene flow analysis to study the evolution of Xa strains from two host plants. Based on the information provided in the paper, few queries need to be answered:

 The sample size of Xa strains collected from different provinces in China is relatively small (39), which may not be representative of the entire population of Xa in China.

·  The authors only used MLSA and rep-PCR fingerprinting to analyze the genetic diversity and population structure of Xa strains, which may not provide a comprehensive understanding of the evolutionary history of this pathogen. Rather, whole-genome sequencing approach will decipher a more comprehensive understanding of the genetic variation and evolution of Xa strains.

·  The authors did not conduct any experimental studies to validate their findings, such as pathogenicity assays or host range tests, which could provide more insights into the biology of Xa strains from different host plants.

 The global distribution and evolution of this pathogen could be achieved with the comparison of Xa populations in China with those in other countries. The authors did not compare their results with other studies on the population structure and genetic evolution of Xa in other countries, which could limit the generalizability of their findings.

Minor revision in language is required.

Author Response

Comment 1: The manuscript deals with study on the population structure and genetic evolution of Xa, which is a major bacterial disease of sugarcane, in China. They collected 39 strains of Xa from six provinces in China and used multilocus sequence analysis (MLSA) and rep-PCR fingerprinting to analyze the genetic diversity and population structure of these strains. The authors also conducted neutral tests, selection pressure analysis, genetic differentiation and gene flow analysis to study the evolution of Xa strains from two host plants. Based on the information provided in the paper, few queries need to be answered:

Reply: Thank you for your comments.

Comment 2: The sample size of Xa strains collected from different provinces in China is relatively small (39), which may not be representative of the entire population of Xa in China.

Reply: I agree with your opinion. The limited numbers of Xa were collected and it may not be representative of the entire population of Xa in China. However, in this manuscript we proposed that some valuable information provide the evidence of obvious genetic divergence and population structure among Xa strains between two host plants (Saccharum spp. hybrid and S. officinarum) from China.

Comment 3: The authors only used MLSA and rep-PCR fingerprinting to analyze the genetic diversity and population structure of Xa strains, which may not provide a comprehensive understanding of the evolutionary history of this pathogen. Rather, whole-genome sequencing approach will decipher a more comprehensive understanding of the genetic variation and evolution of Xa strains.

Reply: In our team we are carrying out the genome and pan-genome sequencing on 39 Xa strains. These results will be published in another manuscript. We mentioned this research gap in the Conclusion section.

Comment 4: The authors did not conduct any experimental studies to validate their findings, such as pathogenicity assays or host range tests, which could provide more insights into the biology of Xa strains from different host plants.

Reply: We have demonstrated that these Xa strains from China were grouped into three classes by pathogenicity assays in a sugarcane cultivar GT58 susceptible to leaf scald. Also, we reported that the cultivar GT58 responses to two Xa strains XaCN51 (high pathogenicity group) and XaCN24 (low pathogenicity group) through a differential modulation of salicylic acid and reactive oxygen species. These results have published in Front. Plant Sci. 2022, 15;13:1087525 (doi: 10.3389/fpls.2022.1087525).

Comment 5: The global distribution and evolution of this pathogen could be achieved with the comparison of Xa populations in China with those in other countries. The authors did not compare their results with other studies on the population structure and genetic evolution of Xa in other countries, which could limit the generalizability of their findings.

Reply: Thank you for your good ideas. This current study focuses on the population structure and genetic evolution of Xa based on the host origins, not on geographic isolation. We mentioned this research gap in the Conclusion section.

Reviewer 2 Report

This article revealed the Genetic Divergence and Population Structure of Xanthomonas albilineans Infecting Sugarcane in China. Before recommending this article for publication, there are some shortcomings for that should be resolve.

The point narrated by the authors in line 12-13 is not suitable. As the evolution and genetic structure is a general mechanism of any species any where. So it would be wrong to say that it is not clear in China. As it is also not clear in other countries.

 So I would recommend to change the statement here and in the title as well.

The authors directly discussed or presented results in the abstract. Methods are not clearly presented in the abstract. As it is essential part of the abstract.

Line 37 should be cited with a recent study. The following study may be helpful. https://doi.org/10.3390/genes13101699

Line 71 which previous studies?. These should be cited.

Main factors of different areas that are affecting the genetic variations and evolution must be discussed.

What are the missing links of this study should be discussed in the discussion section?

 Section 4.1 need to be cited with recent study. https://doi.org/10.1007/s10725-021-00785-7

“The DnaSP v5.0 software was” provide the link.

Line 264-267. the sentence is not clear.

Conclusion looks like summary of the results. Add future recommendations and research gap for future studies.

Carefully read sentences of methodology and revise the sentences which are not clear

Author Response

Comment 1: This article revealed the Genetic Divergence and Population Structure of Xanthomonas albilineans Infecting Sugarcane in China. Before recommending this article for publication, there are some shortcomings for that should be resolve.

Reply: Thank you for your comments.

Comment 2: The point narrated by the authors in line 12-13 is not suitable. As the evolution and genetic structure is a general mechanism of any species any where. So it would be wrong to say that it is not clear in China. As it is also not clear in other countries.

So I would recommend to change the statement here and in the title as well.

Reply: This is sentence was revised as “Little is known about the population structure and genetic evolution of this pathogen.” The title was revised as “Genetic Divergence and Population Structure of Xanthomonas albilineans Strains Infecting Saccharum spp. hybrid and S. officinarum”

Comment 3: The authors directly discussed or presented results in the abstract. Methods are not clearly presented in the abstract. As it is essential part of the abstract.

Reply: Thank you for your comments. We think the current shape of the abstract is OK.

Comment 4: Line 37 should be cited with a recent study. The following study may be helpful. https://doi.org/10.3390/genes13101699

Reply: This sentence is not important content in the text. Here do not need to cite more references.

Comment 5: Line 71 which previous studies?. These should be cited.

Reply: We inserted the three cited references ([7,20,21]) in the text.

Comment 6: Main factors of different areas that are affecting the genetic variations and evolution must be discussed.

Reply: We discussed this issue in the text. In addition to host isolation, some factors affecting the genetic variations and evolution. For example, ecological conditions like soil, climate, water, and other factors differ by geographical locations where sugarcane grows and these conditions can also promote genetic differentiation of pathogen populations.

Comment 7: What are the missing links of this study should be discussed in the discussion section?

Reply: We addressed this issue in the text. For instance, the global distribution and evolution of this pathogen could be revealed by comparing Xa populations in China with those in other countries. A whole-genome sequencing approach should be used to gain a more comprehensive understanding of sequence diversity, virulence, and plant–pathogen interactions. Virulence factors such as effectors in diverse secretion systems in Xa strains might reveal host specificity as well as whether emerging strains are evolving and whether evasion of host immunity systems is occurring.

Comment 8: Section 4.1 need to be cited with recent study. https://doi.org/10.1007/s10725-021-00785-7

Reply: This reference “The Solanum melongena COP1LIKE manipulates fruit ripening and flowering time in tomato (Solanum lycopersicum)” is not associated with our study. We have to say sorry to exclude it in the list of Reference.

Comment 9: “The DnaSP v5.0 software was” provide the link.

Reply: We inserted the related reference “Librado, P., and Rozas, J. (2009). DnaSP v5: a software for comprehensive analysis of DNA polymorphism data. Bioinformatics 25, 1451–1452. doi: 10.1093/bioinformatics/btp187”.

Comment 10: Line 264-267. the sentence is not clear.

Reply: We depicted more about the meanings of parameters of population construct. Negative, neutral and positive selection were indicated by dN/dS ratios < 1, = 1, and > 1, respectively. |Fst| > 0.33 or |Nm| < 1 indicates infrequent gene flow and |Fst| < 0.33 or |Nm| > 1 indicates frequent gene flow.

Comment 11: Conclusion looks like summary of the results. Add future recommendations and research gap for future studies.

Reply: We improved it based on your suggestions and added some research gaps for future studies.

Reviewer 3 Report

Genetic Divergence and Population Structure of Xanthomonas albilineans Infecting Sugarcane in China

By Hu Zhong-Ting, Mbuya Sylvain Ntambo, Jian-Ying Zhao, Talha Javed, Shi Yang, Hua-Ying Fu, Mei-Ting Huang, and San-Ji Gao

The article discusses the population structure and genetic divergence of Xanthomonas albilineans, a leaf scald-causing pathogen of sugarcane. 39 strains of the bacterial species were collected from six different provinces of China and classified based on multilocus sequence analysis, with five housekeeping genes, followed by the use of different fingerprinting tools to estimate the population structure and divergence of Xanthomonas albilineans.

An interesting article, Xanthomonas- Diversity, virulence, and plant-pathogen interactions (Timilsina and coworkers 2020, Nature Reviews Microbiology) provide interesting insights in this direction.

In line 13, it is mentioned that 39 strains of Xa were collected from six provinces in 14 China while in line 17, Among the 93 Xa strains is written, kindly correct it. In addition, 39 strains of Xa were analyzed in the study, what was the rationale for this? Were only 39 strains of Xa found as pathogens in different Chinese provinces? Discuss.

What is the importance/novelty of the work in terms of plant-pathogen interactions?

The English language needs extensive revision for typological errors, like in line 72, the population divergence and involution of this pathogen…..

Is the word evolution or involution?

Line 76, Saccharum spp. Hybird, change to Saccharum spp. Hybrid.

In the Conclusion section, a brief discussion of the study was made. What was the outcome of the study and how does it prove significant for future research in this direction? Discuss in detail.

 The reference section needs to be improved.

The paper needs to be extensively and critically revised for English language/typological errors by an English expert prior to being considered for publication.

Author Response

Comment 1: The article discusses the population structure and genetic divergence of Xanthomonas albilineans, a leaf scald-causing pathogen of sugarcane. 39 strains of the bacterial species were collected from six different provinces of China and classified based on multilocus sequence analysis, with five housekeeping genes, followed by the use of different fingerprinting tools to estimate the population structure and divergence of Xanthomonas albilineans.

Reply: Thank you for your comments.

Comment 2: An interesting article, Xanthomonas- Diversity, virulence, and plant-pathogen interactions (Timilsina and coworkers 2020, Nature Reviews Microbiology) provide interesting insights in this direction.

Reply: This is an excellent review paper. We have cited some insights in the text.

Comment 3: In line 13, it is mentioned that 39 strains of Xa were collected from six provinces in 14 China while in line 17, Among the 93 Xa strains is written, kindly correct it. In addition, 39 strains of Xa were analyzed in the study, what was the rationale for this? Were only 39 strains of Xa found as pathogens in different Chinese provinces? Discuss.

Reply: We have revised the number of “93” as “39”. Leaf scald is just occasionally occurred in sugarcane-planting arrases in China. Thus, a total 39 strains of Xa were isolated in our team up to date. These strains were distributed in six provinces in China.

Comment 4: What is the importance/novelty of the work in terms of plant-pathogen interactions?

Reply: Our work provides the valuable information for further study of interaction between host sugarcane and this pathogen. For instance, investigating the effectors related to secretion system, cell-wall degradation, lipopolysaccharides, and TonB-dependent receptors influencing host specificity and bacterial pathogenicity in this pathogen.

Comment 5: The English language needs extensive revision for typological errors, like in line 72, the population divergence and involution of this pathogen…..

Is the word evolution or involution?

Reply: We have revised the word of “involution” as “evolution”. Also, we have carefully checked these typos.

Comment 6: Line 76, Saccharum spp. Hybird, change to Saccharum spp. Hybrid.

Reply: Revised as your required.

Comment 7: In the Conclusion section, a brief discussion of the study was made. What was the outcome of the study and how does it prove significant for future research in this direction? Discuss in detail.

Reply: We added some research gaps that can be performed in the future work.

Comment 8: The reference section needs to be improved.

Reply: We have carefully improved each cited references according to this journal instructions.

Round 2

Reviewer 3 Report

The authors have taken all suggestions into account, the paper can be accepted in the present form.